# Effect of accentuated eccentric loading countermovement jumps and drop jump training with ladder training versus ladder training alone on sprint performance and change of direction ability in futsal players: A randomized controlled trial protocol

Darpan Chaudhari *, Swapnil U. Ramteke *

Department of Sports Physiotherapy, Ravi Nair Physiotherapy College, Datta Meghe Institute Higher Education and Research (DU), Sawangi (Meghe),Wardha, Maharashtra, India

* dncjal@gmail.com (DC); swapnil.ramteke@dmiher.edu.in (SUR)

## Abstract

Futsal is a fast-paced, high-intensity 5-a-side sport that demands rapid sprints and frequent changes of direction (COD), critical for match performance. While ladder training is known to enhance agility and coordination, combined effects of accentuated eccentric loading (AEL), countermovement jumps (CMJ), and drop jumps (DJ) increase lower body power, it is unclear how these exercises work together to produce futsal-specific results. This paper presents the study protocol for a randomized controlled trial investigating the impact of AEL CMJ, DJ, and ladder training on sprint and COD performance in futsal players. This 6-week, parallel, single-blinded randomized controlled trial, in which outcome assessor will be blinded to group allocation. A total of 62 recreational and competitive futsal players (aged 18–30 years) from futsal turfs across Sawangi Meghe, Wardha. Participants will be randomized (1:1) to receive AEL CMJ, DJ, and ladder training or ladder training alone, 3 times per week. The primary outcomes are the between-group differences in sprint performance (30-meter Sprint Test) and COD ability (Agility T-Test) from baseline to post-intervention (week 6). In order to compare changes between time points and groups, Primary analysis will be conducted using linear mixed-effects models with participant-level random intercepts, following the intention-to-treat principle (CTRI/2025/04/085611).

## Introduction

Futsal is a fast-paced form of small-sided football officially recognized by FIFA. Sport that requires rapid sprints and frequent changes of direction (COD), with 5 players in a team with unlimited substitutions allowed during competitions. As a result, the game's physical demands may be extremely high. Researchers' interest in futsal

**Data availability statement:** No datasets were generated or analysed during the current study, as this manuscript reports a study protocol. Upon completion of the trial and publication of the final results, the fully de-identified individual participant dataset and statistical analysis scripts will be deposited in the Figshare repository and made publicly available without restriction. A permanent digital object identifier (DOI) will be assigned at the time of publication.

**Funding:** The author(s) received no specific funding for this work.

**Competing interests:** I have read the journal's policy, and the authors of this manuscript declare that no authors have competing interests. This does not alter our adherence to PLOS ONE policies on sharing data and materials.

has increased in recent years [1]. During in-play time, players cover over 130 m per minute, totalling over 3300 m every match, with more than 7.5% covered at high intensity (15.5 to 18.3 km·h). Compared to football, it is observed that futsal players change direction more frequently throughout a game while dribbling a ball across a reduced court size. For these reasons, COD may be a particularly relevant physical fitness criterion in futsal [2]. From a contemporary biomechanical perspective, sports characterized by frequent accelerations, decelerations, and rapid stretch–shortening cycle actions—such as futsal—require training strategies that concurrently enhance performance and reduce injury risk by improving neuromuscular control, eccentric load tolerance, and movement efficiency. Recent advances in sports biomechanics emphasize the integration of performance optimization and injury mitigation through evidence-based mechanical loading strategies [3]. To meet these high-intensity demands, training methods must not only improve coordination but also target eccentric muscle strength, power and shortening cycle (SSC) efficiency. Ladder drills are commonly used to enhance footwork, coordination, agility and reaction time. Providing structured environment for practicing multidirectional movement [4,5]. However, ladder drills primarily stimulate low-intensity, coordination-focused neuromuscular activity, providing limited eccentric or plyometric load, which may be insufficient for maximizing futsal-specific power and speed.

To address this limitation, Experts in Research and practitioners have incorporated eccentric-based training to properly load the eccentric motion by removing the restriction of concentric force output. The magnitude of the mechanical stimulation usually determines the skeletal muscle response, and it has been demonstrated that eccentric-only training increases response, particularly in terms of strength and size changes [6]. Eccentric-only training most commonly via the Nordic Hamstring Exercise has been shown to influence sprint performance in football/soccer athletes, with randomized trials and reviews reporting small-to-moderate improvements in short-distance sprint times, alongside increases in eccentric hamstring strength and fascicle length [7,8].

Recently, increased emphasis has been placed on sports performance and the application of accentuated eccentric loading (AEL). AEL involves using eccentric and concentric contractions to administer an eccentric load greater than the necessary concentric load while attempting to preserve natural movement patterns [9]. Recent systematic evidence indicates that eccentric resistance training protocols can produce greater improvements in muscle strength and functional capacity compared with traditional resistance training, providing a stronger empirical basis for incorporating eccentric-focused methods such as AEL into sport-specific conditioning programs [10]. For instance, a coach may load a back squat with a specific weight for the eccentric section and then manually remove the weight before beginning the concentric action. This strategy is thought to improve adaptation by increasing eccentric loading, which results in more eccentric and concentric force output. Evidence shows that this training strategy causes shifts to quicker myosin heavy chain (MHC) isoforms and more favourable modifications in bispecific muscle [11]. Increases in force and power production have frequently accompanied these modifications.

Furthermore, past research reveals beneficial changes in jumping and throwing movements, indicating AEL may transfer well to sport demands and performance when applied to plyometric training exercises [12].

Furthermore, chronic AEL CMJ training has regularly exceeded traditional CMJ training in developing lower body power (+20% vs. + 1%) and vertical jump height (+11% vs. −2% over time) [13]. To our knowledge, no research has established if AEL applied to a CMJ can increase other independent measures of physical performance, such as lower body strength, sprinting, and cognitive performance. The drop jump (DJ) involves stepping off a raised platform and leaping right off the ground. The countermovement leap (CMJ), on the other hand, starts from a standing position and accelerates explosively upward after a downward movement. According to earlier studies, the DJ can significantly produce higher force (+2.3 to 31.7%), power production (+0.8%), and jump height (+6.4 to 13.2%) than the CMJ [14]. With these advantages, DJ training has effectively increased sprint performance, lower body strength, vertical jump height, lower body power, and change of direction (COD) ability. This enhancement is attributed to the DJ increased eccentric loading phase, which enhances motor unit recruitment and activation and potential kinetic energy storage and usage [15]. Although both AEL CMJ and drop jumps target the stretch–shortening cycle, they emphasize distinct neuromuscular qualities. AEL CMJ primarily overloads the eccentric phase and enhances force production during longer SSC actions, whereas drop jumps preferentially train fast SSC behavior characterized by minimal ground contact time and high reactive strength. The combined application of AEL CMJ and DJ therefore allows comprehensive SSC development, addressing both braking force capacity and rapid re-acceleration demands inherent to sprinting and change-of-direction tasks in futsal [8]. However, improvements in force and SSC efficiency alone may not fully translate to sport-specific movement unless integrated with coordination- and pattern-based drills that replicate futsal movement demands the agility ladder is a reasonably priced and simple training aid that enables coaches and athletes to be creative in adjusting task restrictions during drills and cultivating the movement coordination patterns common in team sports. Ladder workouts are designed to enhance and improve athletic performance and footwork. It improves foot speed, mobility, proprioception, agility, power, strength, balance, coordination, and response time [5]. Because ladder drills encourage the body and mind to synchronise different foot movements, training sessions are more engaging when conducted rhythmically. Depending on the training objective, a ladder has two straps and rungs spaced 15–18 inches apart. Like an agility ladder, a ladder can be constructed at home with PVC pipe and rope, then taped to the floor. Ladder training helps players become more coordinated and catch, hit, block, and tackle more easily. Leaps are performed swiftly, in various directions, and without any obstacles on the agility ladder. Running through a ladder, skips, shuffles, jumps/hops, and linear and lateral movements are the four fundamental skills of ladder training [16]. We hypothesize that combining accentuated eccentric loading countermovement jumps and drop jumps with ladder drills will lead to greater improvements in sprint speed and change-of-direction performance than ladder drills alone.

## Methods

### Study design

The experimental group will participate in a 6-week, three-weekly, 45-minute session that includes drop jump with ladder training and accentuated eccentric loading of CMJ. This study will be a parallel-group randomized controlled trial. Ladder training will be given to the control group. Assessments will be finished at baseline (week 0) and post-intervention (week 6). The CONSORT (Consolidated Standards of Reporting Trials) declaration will be followed when reporting quantitative trial results (Figs 1 and 2)

### Participants and study setting

Participants for the study will be recruited through Flyers and study announcements will be distributed to futsal clubs and recreational futsal groups affiliated with the Ravi Nair Physiotherapy College (RNPC) Department of Sports

| | | Study period | | | | | | | | |
|---|---|---|---|---|---|---|---|---|---|---|
| | Enrolment | Allocation | Post-allocation | | | | | | Close-out | |
| Timepoint** | -t₁ | 0 | T₁ | T₂ | T₃ | T₄ | T₅ | T₆ | Tₓ | |
| **Enrolment:** | | | | | | | | | | |
| Eligibility screen | X | | | | | | | | | |
| Informed consent | X | | | | | | | | | |
| *[list other procedures]* | X | | | | | | | | | |
| Allocation | | X | | | | | | | | |
| **Interventions:** | | | | | | | | | | |
| Group A: AEL CMJ + DJ + Ladder | | | ◆————————————————◆ | | | | | | | |
| Group B: Ladder Training | | | ◆————————————————◆ | | | | | | | |
| **Assessments:** | | | | | | | | | | |
| Baseline outcomes (30-m Sprint, Agility T-Test) | X | | | | | | | | | |
| *Training adherence and adverse event* | | | X | X | X | X | X | X | X | |

**Fig 1. SPIRIT schedule of enrolment, interventions, and assessments across the 6-weeks study period.**

Physiotherapy, A written consent form and an information sheet will be given to interested futsal players at an introductory session regarding the project. The RNPC Department of Sports Physiotherapy is organizing the study, which will be carried out at futsal turfs throughout Sawangi meghe, Wardha. Futsal players who fit the following requirements will be invited to take part in the study. Written consent form and an information sheet will be given to interested futsal players at an introductory session regarding the research. Recruitment has not yet begun; it is scheduled to start on 01/07/2025 and will continue until 30/09/2025. The RNPC Department of Sports Physiotherapy will coordinate the study, which will be carried out at futsal turfs throughout Sawangi meghe. Participants in the study will be futsal players who fulfill the following qualifying requirements.

**Eligibility criteria.** *Inclusion criteria:*

Age is taken between 18–30 years

BMI-18.5–24.9

Regular participation in futsal activities since 1 year, defined as ≥2 sessions per week, each lasting ≥60 minutes.

Recreational and competitive futsal players participating in regional, national, or professional leagues

Willingness to participate in a structured 6-week intervention program.

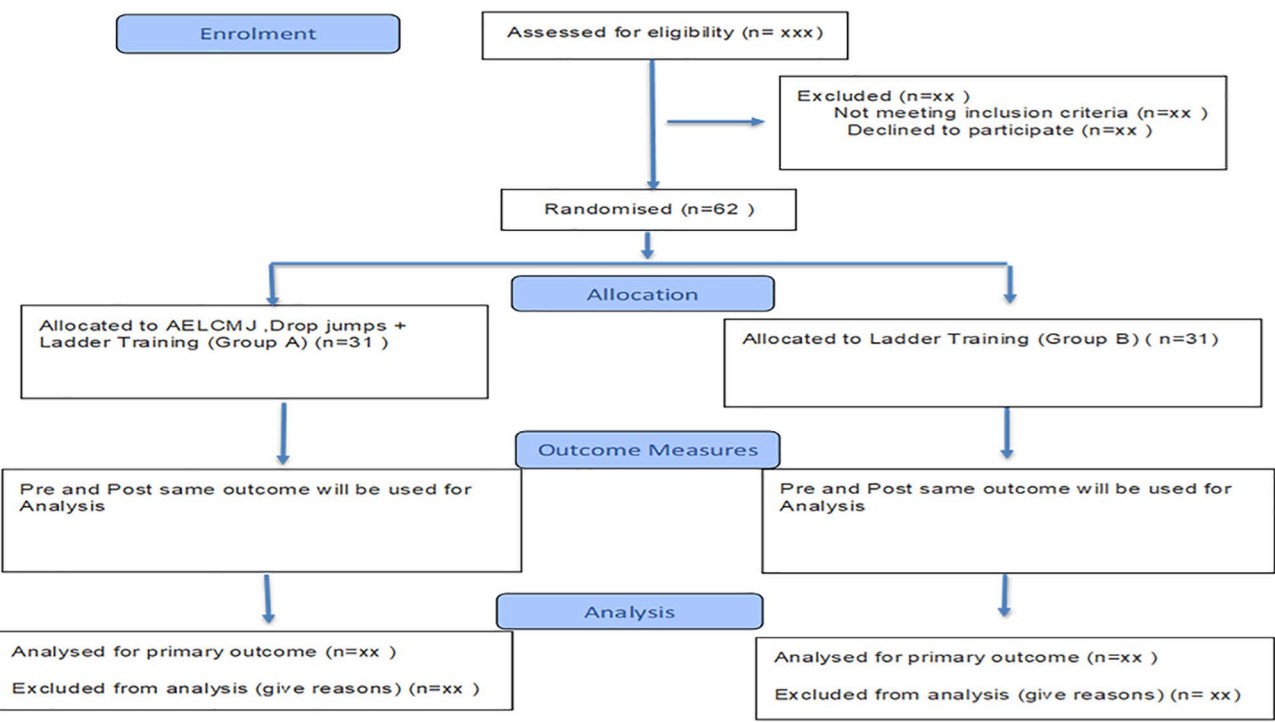

**Fig 2. CONSORT diagram of the trials showing participant progress through enrollment, allocation, intervention, outcome measures, and analysis.**

*Exclusion criteria:*

presently having pain due to trauma associated to lower limb

laxity of articular ligaments or meniscus injury around knee

recent history of dislocation of patella or recent fracture

chondromalacia patella

existing progressive neurological conditions

plantar fasciitis

### Randomization and blinding

Eligible participants who complete baseline measurements will be randomly assigned to either the experimental group (AEL CMJ+DJ+ladder training) or the control group (ladder training only) using a web-based computer-generated randomization program with permuted blocks. Block sizes will vary between 4 and 6, and randomisation will be stratified by sex(male/female) and competition level (recreational / competitive). Allocation will be prepared in advance by an independent researcher not involved in participant recruitment, intervention delivery, or outcome assessment. Allocation codes will be stored in sequentially numbered, opaque, sealed envelopes. After baseline evaluation, the trial coordinator will obtain the allocation from the independent researcher and assign participants accordinglyBlinding: Outcome assessors and data analysts will be blinded to group allocation to minimize assessment and analysis bias. Due to the nature of the intervention, participants and trainers cannot be blinded to group allocation; however, participants will not be informed of the study hypotheses to reduce expectation bias. Research staff responsible for data collection, entry, and analysis will remain blinded throughout the trial.

## Interventions

Groups A and B will each receive a 6-week intervention that consists of 3 x 45-minute sessions each week. Both groups will get standard physiotherapy (Ladder Training) treatment. Along with ladder training, Group A will also receive drop jumps and accentuated eccentric loading (AEL) countermovement jumps, while Group B will only receive ladder training.

Experimental Group-A: Accentuated eccentric loading Countermovement jump, Drop jumps with Ladder training

Group A participants will receive Accentuated eccentric loading Countermovement jumps, drop jumps with Ladder training alongside Ladder training to improve sprint performance. Over 6 weeks, participants will progressively increase intensity, starting with 4 sets of 8 repetitions and progressing to 4 sets of 10 repetitions. AEL CMJs will be performed using weighted dumbbells or a harness (10–20% body mass) during the eccentric phase, with the load released prior to the concentric jump. Load progression is individualized and guided by proper technique, participant-reported RPE ≤ 6/10, and physiotherapist assessment following established plyometric training guidelines to ensure progressive overload, neuro-muscular adaptation, and safety. Drop jump height will start at 30 cm and may progress up to 50 cm, contingent on stable landing mechanics, absence of knee valgus, and RPE ≤ 6/10, with coaches monitoring safety and adjusting individually. Sessions will be structured as 4 sets × 8 reps per exercise during weeks 1–3 (~32 contacts) and 4 sets × 10 reps during weeks 4–6 (~40 contacts). Total ground contacts per session are matched (≈100). Total ground contacts per session were numerically matched (≈100) between groups. Contact counts were recorded using a manual click counter and verified in 10% of sessions via video review to ensure consistency of session structure. It is acknowledged that the mechanical and neuromuscular load of AEL CMJs and drop jumps is substantially higher than ladder drills. Therefore, this study compares the type of loading rather than the absolute intensity between groups, and the results should be interpreted with this limitation in mind. Contacts were tallied during each session using a manual click counter by the supervising coach and verified for 10% of sessions via video review to ensure accuracy and training-volume equivalence between groups. with Group B to eliminate training-volume confounding (Table 1). Intervention fidelity was monitored using a standardized checklist (S5 File), which tracked completion of each exercise, technique quality, set/repetition adherence, AEL load progression, and total ground contacts. 10 percent of sessions were also randomly video-verified to ensure adherence to the protocol.

Control Group-B: Ladder Training

Participants in the control group will perform ladder training only over a 6-week period, 3 sessions per week, 45 minutes per session. Ladder drills will progress weekly to increase complexity and coordination: weeks 1–2 include straight runs, hopscotch, and single-foot zigzag hops; weeks 3–4 include two-foot runs, backward hopscotch, and single-foot lateral in/out hops; and weeks 5–6 include bunny hops, hopscotch variations, and two-foot zigzag hops. Total session volume (~100 ground contacts per session) and warm-up/cool-down routines will be matched to the experimental group participants will perform ladder drills at a moderate intensity (RPE 5–6/10), equivalent to the perceived exertion of the experimental group. to ensure that any observed effects are attributable to AEL CMJ and drop jumps rather than differences in training volume or session duration. Although session duration was matched between groups, the interventions were not intended to equate external or internal physiological load. The ladder training protocol was identical in both groups; however, Group B performed additional ladder sets to match session duration, not mechanical loading (Table 2)

## Outcome measures

**Primary outcome measure.** 30 M Sprint Test-

A popular field test for determining a person's linear speed and acceleration over a brief distance is the 30-meter sprint test. The time it takes for participants to run 30 meters at their maximum speed from a stationary starting point is recorded; for better precision, electronic timing gates are usually used. The test is simple, practical, and provides immediate results, making it valuable for talent identification and monitoring speed development in athletes. Research demonstrates that the 30-meter sprint test has excellent reliability, with intraclass correlation coefficients (ICC) ranging from 0.92

**Table 1. Accentuated eccentric loading countermovement jump, drop jump training, with ladder training.**

| Group A: Accentuated eccentric loading countermovement jump, drop jump training, with ladder training | | | | | |
|---|---|---|---|---|---|
| Exercise Category | Procedure | Sets × Reps / Contacts | Duration (min) | Rest | Rationale & Progression |
| Warm-up | Dynamic mobility drills + activation (hip, knee, ankle) | — | 10 min | — | Prepares muscles/joints, increases core temperature, reduces injury risk. Identical for both groups. |
| Accentuated Eccentric Loading (AEL) CMJ | Use dumbbells/harness (10–20% BodyMass) for eccentric phase; drop load before concentric jump. | Weeks 1–3: 4 × 8 (≈32 contacts) Weeks 4–6: 4 × 10 (≈40 contacts) | 5 min | 2 min between sets | Increases eccentric force production, neural drive, and stretch-shortening cycle Load progression is based on proper technique and RPE ≤ 6/10, increasing 2–5% per week to ensure safe progressive overload. |
| Drop Jumps (DJ) | Step off the plyo box, land softly, and immediately rebound vertically. | Weeks 1–3: 4 × 8 (≈32 contacts) Weeks 4–6: 4 × 10 (≈40 contacts) | 5 min | 2 min between sets | Enhances reactive strength and COD ability. Drop height progresses 30→50 cm based on landing mechanics, absence of knee valgus, and RPE ≤ 6/10; physiotherapists monitor safety and adjust individually. |
| Ladder Training | • Weeks 1–2: straight run, hopscotch, single-foot zig-zag hops. • Weeks 3–4: two-foot run, backwards hopscotch, single-foot lateral in/out hops. • Weeks 5–6: bunny hops, hopscotch variations, two-foot zigzag hops. | 6 reps × 2 sets per drill ≈ 36 contacts | 20 min | 30 s between reps; 60 s between sets | Improves foot speed, coordination, and agility. Ladder volume is identical to Group B to ensure fair exposure. |
| Cool-down | Light jogging, static stretching | — | 5 min | — | Promotes recovery, flexibility, and gradual HR reduction. Identical for both groups. |
| Total Session Volume | Ladder + AEL + DJ | ≈ 100 total ground contacts per session | 45 min | — | Session duration and total contacts are matched to Group B to control for training volume. |

**Table 2. Ladder training.**

| Group B: Ladder Training | | | | | |
|---|---|---|---|---|---|
| Exercise Category | Procedure | Sets × Reps / Contacts | Duration (minutes) | Rest | Rationale & Progression |
| Warm-up | Same as Group A | — | 10 min | — | Ensures equivalent preparation between groups. |
| Ladder Training (Full Session) | Same drill sequence and weekly progression as Group A. Extra ladder block added (~28 contacts) to match Group A's total contacts from AEL + DJ | 4 drills × 6 reps × 2 sets ≈ 72 contacts + additional ladder block ≈28 contacts during the 10-min period Group A performs AEL + DJ | 30 min(-main) + 10 min(additional) = 40 total ladder minutes | 30 s between reps; 60 s between sets | The additional ladder block consisted of repetitions of the same ladder drills prescribed for that training week, performed at the same moderate intensity (RPE 5–6/10). This block was included solely to match total session duration and ground contacts with Group A and did not introduce additional plyometric or high-impact loading. |
| Cool-down | Same as Group A | — | 5min | — | Identical recovery protocol. |
| Total Session Volume | Ladder only | ≈ 100 total ground contacts per session | 45min | — | Matched session duration and contacts eliminate training-volume confounder. |

to 0.98 when standardized protocols and timing equipment are used. This high reliability is consistent across different age groups and starting conditions, supporting its use in both youth and adult sports settings [17].
Agility T-Test-

A multidirectional, field-based agility test that gauges a person's quick acceleration, deceleration, and direction changes. The test consists of backpedalling, lateral shuffling, and sprinting around a cone-marked T-shaped course. Scores are recorded in seconds, with faster times indicating better agility performance. The Agility T-Test demonstrates excellent test-retest reliability, with intraclass correlation coefficients (ICC) typically reported between 0.94 and 0.98 [18]. However, while widely used, traditional T-Tests may not fully capture the multidirectional and reactive demands of team sports such as futsal. Recent methodological advancements highlight three-dimensional agility assessments that integrate horizontal, vertical, and sport-specific reactive movements, offering greater ecological validity and specificity for evaluating complex game-related agility [19].

## Sample size calculation

As limited literature exists on the combined effects of accentuated eccentric loading (AEL) and ladder training on sprint performance in futsal players, a pragmatic sample size calculation was undertaken using data from previous plyometric and eccentric overload training studies in trained athletes. The expected mean difference in 30 m sprint time ($\delta = 0.064$ s) was derived from the pre–post changes reported in a randomized controlled trial [15] and supported by similar magnitude sprint improvements reported in high-intensity plyometric training studies in team-sport athletes [20]. The pooled standard deviation ($\sigma = 0.125$ s) was calculated from the baseline and post-intervention standard deviations reported in these studies.

The sample size per group was calculated using the standard formula for comparison of two independent means:

$$n_1 = n_2 = \frac{2(Z_\alpha + Z_\beta)^2 \sigma^2}{\delta^2}$$

Where:

$Z_\alpha$ = 1.96 (two-tailed, $\alpha = 0.05$), $Z_\beta$ = 0.84 (corresponding to 80% statistical power), $\sigma$ = pooled standard deviation, $\delta$ = expected mean difference.

A statistical power of 80% was chosen as it is widely accepted in clinical and sports performance research to balance the risk of Type II error with feasibility in athletic populations. Substituting these values resulted in a minimum required sample size of 28 participants per group. To account for an anticipated 10% dropout rate, the total sample size was increased to 62 participants (31 per group). Although a mean difference of 0.064 seconds may appear small, it is considered clinically meaningful in futsal, where rapid acceleration and first-step quickness over short distances are crucial for winning duels, creating space, and executing successful change-of-direction actions. Previous research has demonstrated that improvements as small as 0.05–0.10 s over 20–30 m can translate into meaningful competitive advantages in elite team-sport performance.

## Data collection, management, and analysis

**Data collection methods.** Pre-intervention data will be collected upon enrolment; additional post-intervention data will be collected after 6 weeks.

**Data management.** Data will be encrypted, stored in a secure cloud database, and de-identified. Hard copies will be locked at RNPC. Only authorized researchers will access data.

**Statistical method.** All randomized participants fulfilling the inclusion and exclusion criteria will be analyzed according to the intention-to-treat (ITT) principle. Participants who withdraw or lack post-intervention outcome data will remain in the ITT analysis, with missing values handled using multiple imputation. Baseline characteristics will be summarized using mean ± standard deviation (SD) for continuous variables and frequency (n) and percentage (%) for categorical variables, while continuous outcomes (30-Meter Sprint Test and Agility T-Test) will be presented with mean, SD, standard error, minimum, maximum, and 95% confidence intervals (CI).

The primary analysis will use a linear mixed-effects model (LMM) to examine the effects of group (AEL + DJ + Ladder vs. Ladder only), time (pre- and post-intervention), and their group × time

interaction on sprint performance and change-of-direction ability. The model will be specified as
$Y_{ij} = \beta_0 + \beta_1 \text{Group}_i + \beta_2 \text{Time}_j + \beta_3 (\text{Group}_i \times \text{Time}_j) + \gamma_1 \text{BaselineOutcome}_i + \gamma_2 \text{Age}_i + u_i + \in_{ij}$, where $Y_{ij}$ represents the outcome for participant $i$ at time $j$, $\beta_0$ is the intercept, $\beta_1, \beta_2, \beta_3$ are fixed effects for group, time, and their interaction, $\gamma_1$ and $\gamma_2$ represent covariates for baseline outcome and age, $u_i$ is the participant-specific random intercept to account for repeated measures, and $\in_{ij}$ is the residual error. Baseline imbalances in key covariates such as age, sex, training experience, or baseline performance will be addressed by including these variables as fixed-effect covariates in the model, ensuring that the adjusted group × time contrasts represent the primary intervention effect.

Effect-size metrics will include adjusted mean differences (least-squares means) with 95% confidence intervals and standardized effect sizes (Cohen's d) where appropriate. To control for multiplicity, a Bonferroni-adjusted significance threshold ($\alpha = 0.025$) will be applied for the two co-primary outcomes. Model assumptions will be assessed using Shapiro–Wilk tests and Q-Q plots for normality and Levene's test for homogeneity of variance, with data transformed or analyzed with non-parametric methods if assumptions are violated. Floor and ceiling effects will also be evaluated, with >15% of participants at the minimum or maximum considered significant [21]. Sensitivity analyses will evaluate the robustness of the findings and will include per-protocol analyses of participants completing ≥80% of attendance and prescribed training volume, complete-case analyses excluding participants with missing outcomes, and alternative analyses using independent-samples t-tests or Mann–Whitney U tests on change scores.

Missing outcome data will be addressed using multiple imputation by chained equations (MICE) under the missing-at-random (MAR) assumption. Twenty imputed datasets will be generated, incorporating group allocation, time point, baseline outcome, age, and other relevant performance-related variables. Analyses will be performed using SPSS version 26.0 (IBM Corp., Armonk, NY, USA), with primary inference based on pooled estimates from the linear mixed-effects model (LMM) across the imputed datasets. Statistical significance will be set at p ≤ 0.05, with a Bonferroni-adjusted threshold of $\alpha = 0.025$ applied for the two co-primary outcomes.

## Monitoring

**Data monitoring.** The Data Monitoring Committee of Ravi Nair Physiotherapy College will oversee trial conduct and participant safety. Safety monitoring will be performed throughout the intervention period, with formal checks conducted weekly and after each training session. Attendance, training load, and any adverse events will be recorded by the supervising physiotherapist. No formal interim efficacy analysis is planned due to the short duration of the intervention.

**Harms.** Given the higher eccentric loading involved in accentuated eccentric loading countermovement jumps and drop jumps, all training sessions will be closely supervised by qualified physiotherapists. Any adverse events, including musculoskeletal pain, strain, or injury, will be documented using a standardized adverse event reporting form.

Adverse events will be classified as mild, moderate, or severe by the clinician in charge. In the event of a serious adverse event or if more than two participants experience similar moderate adverse events related to the intervention, training will be paused and the Institutional Ethics Committee will be informed immediately. Decisions regarding continuation, modification, or termination of the trial will be taken jointly by the Ethics Committee and the principal investigator.

Participants experiencing adverse events will receive appropriate medical care through institutional facilities, and compensation will be provided as per institutional ethics committee and university policies. All adverse events and harms will be reported in the final trial publication in accordance with CONSORT and PLOS ONE reporting guidelines.

## Ethics and dissemination

**Research ethics approval.** This Protocol has been registered with Clinical Trial Registry-India (CTRI): CTRI/2025/04/085611, registered on 24/04/2025, URL of the CTRI: https://ctri.nic.in/Clinicaltrials/regtrial. php?modid=1&compid=19&EncHid=56210.33725. The Study Trial has been approved by the Datta Meghe Institute of

Higher Education and Research(Deemed to Be University), Re-Accredited with NAAC Grade A++, Institutional ethics committee IEC approval number: DMIHER(DU)/IEC/2025/611 S3 File.

**Consent.** Participants will be thoroughly informed about the study objectives, procedures, potential risks, and benefits. Written informed consent will be obtained from all participants prior to enrollment. Signed consent forms will be securely stored by the principal investigator at the Department of Physiotherapy, Datta Meghe Institute of Higher Education and Research, for a minimum period of five years after study completion, in accordance with institutional and ethical guidelines.

**Confidentiality.** All data pertaining to research participants will be kept private. Only with the participants' express consent will patient-related data be used.

**Data access.** Data will be securely stored under the supervision of the chief investigator, in accordance with institutional ethics approval. Upon publication of the final manuscript, the fully de-identified individual participant dataset and accompanying statistical analysis scripts will be deposited in the Figshare repository and made publicly available. A permanent digital object identifier (DOI) will be assigned at the time of publication to ensure accessibility and traceability.

**Dissemination.** Information gathered during or following only academic and research-related goals that lead to a publication will be pursued by the study in a reputable journal.

## Discussion

Futsal's high-intensity demands, including frequent sprints and changes of direction (COD), necessitate targeted training to optimize player performance [22]. This randomised controlled trial (RCT) protocol evaluates the efficacy of combining AEL CMJ, DJ, and ladder training, compared to ladder training alone, on sprint performance and COD ability in futsal players. By addressing the scarcity of evidence on AEL's sport-specific benefits, this study aims to advance training methodologies for futsal athletes.

Prior research has established AEL's superiority over traditional plyometric training for enhancing lower body power and vertical jump height, yet its effects on sprint and COD performance remain underexplored, particularly in futsal [23]. Existing studies often focus on non-athletic populations or vertical jump outcomes, with small sample sizes limiting generalizability to sport-specific contexts. This RCT builds on these findings by integrating AEL CMJ, DJ, and ladder drills, leveraging the stretch-shortening cycle and coordination to target futsal-specific demands [24]. The intervention could provide coaches and physiotherapists with a superior method to improve sprint speed and agility, enhancing match performance like ball recovery and defensive maneuvers. The findings may also inform rehabilitation protocols for futsal players recovering from lower limb injuries, particularly by supporting modern joint-by-joint training approaches that emphasize coordinated strength and mobility across the kinetic chain rather than focusing solely on isolated knee exercises [25]. This paradigm may help optimize lower-limb mechanics, reduce injury risk, and complement sport-specific conditioning programs.The trial's strengths include randomization via a computer-generated sequence, minimizing allocation bias, and using validated outcomes—the 30-meter Sprint Test and Agility T-Test—ensuring sport-specific relevance [18]. Adherence to SPIRIT 2013 guidelines and registration with CTRI enhances transparency [26].If this RCT confirms our hypothesis, integrating accentuated eccentric loading countermovement jumps and drop jumps with ladder drills is expected to produce greater improvements in sprint and change-of-direction performance in futsal players. These findings could shape conditioning and rehabilitation programs and deepen understanding of how eccentric overload and stretch-shortening cycle efficiency improve sport-specific performance. Ladder training is well-established to improve change-of-direction (COD) ability and agility in futsal players. In this trial, the experimental group combines AEL CMJs and drop jumps with ladder training to investigate whether a higher mechanical and neuromuscular load can produce equal or superior improvements in COD and sprint performance. A key limitation of this study is the intensity mismatch between the experimental (AEL CMJ+DJ+Ladder) and control (Ladder only) groups. Although total ground contacts were matched (~100 per session), the external mechanical load and internal neuromuscular demand associated with AEL CMJs and drop jumps

are substantially higher than those of ladder drills, which primarily emphasize coordination and footwork. Consequently, any observed between-group differences may reflect not only the type of training but also the magnitude of loading. This limitation should be considered when interpreting the results, and future studies may incorporate low-intensity plyometric exercises in control groups to better match neuromuscular stimulus while isolating the effects of eccentric overload.

## Supporting information

**S1 File. Cover letter.** https://figshare.com/s/0d04625263b02b203d2a.
(DOCX)

**S2 File. Spirit checklist.** https://figshare.com/s/a3c68cb1f34b4a737b3d.
(PDF)

**S3 File. Institutional ethical committee approval.** https://figshare.com/s/1cf7a70e82833904420d.
(PDF)

**S4 File. Original protocol sent to ethics committee. (English).** https://figshare.com/s/165de7842907f9361f86.
(PDF)

**S5 File. Fidelity checklist.** https://figshare.com/s/068bc0d3674d130cb312.
(PDF)

## Acknowledgments

The author expresses gratitude in this part for the critical assistance provided by certain colleagues, institutions

## Author contributions

**Conceptualization:** Swapnil U. Ramteke.

**Methodology:** Darpan Chaudhari.

**Supervision:** Swapnil U. Ramteke.

**Validation:** Swapnil U. Ramteke.

**Visualization:** Swapnil U. Ramteke.

**Writing – original draft:** Darpan Chaudhari.

**Writing – review & editing:** Darpan Chaudhari.

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
