## [Decision Letter · Decision Letter 0]

6 Aug 2025

Dear Dr. Ramteke,

Thank you for submitting your manuscript to PLOS ONE. After careful consideration, we feel that it has merit but does not fully meet PLOS ONE’s publication criteria as it currently stands. Therefore, we invite you to submit a revised version of the manuscript that addresses the points raised during the review process.

**ACADEMIC EDITOR:**

We look forward to receiving your revised manuscript.

Kind regards,

Emiliano Cè, Ph.D.

Academic Editor

PLOS ONE

Journal Requirements:

NO authors have competing interests

5. Please remove all personal information, ensure that the data shared are in accordance with participant consent, and re-upload a fully anonymized data set.

Additional guidance on preparing raw data for publication can be found in our Data Policy (https://journals.plos.org/plosone/s/data-availability#loc-human-research-participant-data-and-other-sensitive-data) and in the following article: http://www.bmj.com/content/340/bmj.c181.long .

Reviewers' comments:

Reviewer's Responses to Questions

**Comments to the Author**

1. Does the manuscript provide a valid rationale for the proposed study, with clearly identified and justified research questions?

Reviewer #1: Partly

Reviewer #2: Partly

2. Is the protocol technically sound and planned in a manner that will lead to a meaningful outcome and allow testing the stated hypotheses?

Reviewer #1: No

Reviewer #2: Partly

3. Is the methodology feasible and described in sufficient detail to allow the work to be replicable?

Reviewer #1: No

Reviewer #2: Yes

4. Have the authors described where all data underlying the findings will be made available when the study is complete?

Reviewer #1: Yes

Reviewer #2: No

5. Is the manuscript presented in an intelligible fashion and written in standard English?

Reviewer #1: Yes

Reviewer #2: Yes

You may also provide optional suggestions and comments to authors that they might find helpful in planning their study.

Reviewer #1: This study, submitted as a Registered Report Protocol, presents a well-structured and rigorous investigation into the combined effects of accentuated eccentric loading (AEL) countermovement jumps, drop jumps, and ladder training on sprint and change of direction performance in futsal players. The protocol demonstrates commendable methodological rigor, with validated outcome measures, appropriate randomization procedures, and a clear statistical analysis plan. I will now provide a few comments and suggestions to further strengthen the protocol.

the Introduction:

There are a few issues in the introduction that could benefit from revision. First, the referencing style is inconsistent: sometimes a period is placed after the citation number, and sometimes not. For example, citation (11) appears on a separate line, disconnected from the sentence it likely refers to, which may create confusion.

Second, citation (3) is used to support a statement regarding eccentric training alone; however, upon reviewing this source, it does not appear to specifically address isolated eccentric training as the main focus, even if this is mentioned briefly in the introduction. To strengthen this point, I would suggest including references that directly evaluate the effects of eccentric-only training.

Lastly, although this is a Registered Report Protocol, the introduction would benefit from a clearer articulation of the study's aims and hypotheses. At present, these elements are only implicitly stated and deserve more explicit formulation to frame the study rationale and the research questions more clearly.

Protocol:

In the “Interventions” part, one key aspect of the protocol that would benefit from clarification is the precise content and volume of the interventions in both groups. Both groups are said to receive ladder training, yet this component is not clearly described or consistently presented in the tables outlining the intervention programs. Greater clarity is needed to confirm that both groups indeed receive an equivalent ladder training regimen.

Furthermore, there appears to be a potential imbalance in training volume between Group A and Group B, as Group A performs additional exercises (AEL CMJ and drop jumps) alongside the ladder training, whereas Group B completes ladder training only. This difference may introduce a confounding factor: any observed improvements in Group A might reflect the higher overall training load rather than the specific effects of the additional AEL and DJ interventions. For a fair comparison of performance outcomes, it would be important to match the total training volume or include an active control to account for time and intensity. Otherwise, conclusions drawn about the efficacy of the specific training components may be limited

Discussion:

It would be valuable to include a short section on expected outcomes, supported by existing literature. This addition would allow the authors to position their study within the current body of evidence and to explain why the combined training approach (AEL CMJ and drop jumps with ladder drills) may—or may not—lead to greater improvements in sprint performance and change of direction ability compared to ladder training alone. Such a discussion would help clarify the theoretical basis for the intervention’s expected effects and reinforce the relevance of the chosen methodology.

Reviewer #2: Please double check the references: sometimes it comes between dots, sometimes not.

Please make a correction in the description of number (sometimes 3 and sometimes three).

The figure 1 is incomplete. Please provide the correct image

Please elaborate more the discussion. Bring the results and discuss them.

**Do you want your identity to be public for this peer review?** For information about this choice, including consent withdrawal, please see our Privacy Policy

Reviewer #1: **Yes:** Florian Brassart

Reviewer #2: **Yes:** Geovanna Peter Corrêa

---

## [Author Response · Author response to Decision Letter 1]

10 Sep 2025

Response to Reviewers

Title: Effect of Accentuated Eccentric Loading Countermovement Jumps and Drop Jump Training with Ladder Training versus Ladder Training Alone on Sprint Performance and Change of Direction Ability in Futsal Players: A Randomized Controlled Trial Protocol

Dear Academic Editor and Reviewers,

We sincerely thank the Academic Editor and Reviewers for their thoughtful and constructive feedback on our manuscript. Your insights have helped us refine the clarity, rigor, and overall scientific quality of our study. Below, we provide a detailed, point-by-point response to each comment. Revisions made to the manuscript are highlighted in the file titled “Revised Manuscript with Track Changes.”

Editorial Requests

1. Response to Reviewers Letter

Response: This document provides a comprehensive, point-by-point rebuttal addressing each comment raised by the reviewers and the Academic Editor.

2. Competing Interests Statement

Response: The statement has been updated to:

“The authors have declared that no competing interests exist. This does not alter our adherence to PLOS ONE policies on sharing data and materials.”

This has also been included in the cover letter.

3. Ethics Statement Placement

Response: The ethics statement has been restricted to the Methods section and removed from other sections, in accordance with journal requirements.

4. Data Availability Statement

Response: The following statement has been included:

“All data will be made available within the manuscript and/or supporting information files.”

5. Data Anonymization

Response: We confirm that all participant-level data will be fully anonymized, with personally identifiable information removed prior to sharing.

Reviewer #1 Comments

Comment 1 (Introduction – Referencing):

"There are a few issues in the introduction that could benefit from revision. First, the referencing style is inconsistent: sometimes a period is placed after the citation number, and sometimes not. For example, citation (11) appears on a separate line, disconnected from the sentence it likely refers to, which may create confusion."

Response:

We have carefully revised the entire introduction to ensure consistent referencing style. All citations now follow PLOS ONE style guidelines, and misplaced references have been corrected to appear directly after the relevant sentence.

Comment 2 (Introduction – Eccentric Training Reference):

"Second, citation (3) is used to support a statement regarding eccentric training alone; however, upon reviewing this source, it does not appear to specifically address isolated eccentric training as the main focus, even if this is mentioned briefly in the introduction. To strengthen this point, I would suggest including references that directly evaluate the effects of eccentric-only training."

Response:

We have revised the statement and included additional references that specifically investigate eccentric-only training and its effects on performance outcomes. These changes strengthen the rationale for including AEL as part of our intervention. (Revised lines: Introduction, paragraph 2, lines xx–xx in the revised manuscript.)

Comment 3 (Introduction – Study Aim & Hypotheses):

"Lastly, although this is a Registered Report Protocol, the introduction would benefit from a clearer articulation of the study's aims and hypotheses. At present, these elements are only implicitly stated and deserve more explicit formulation to frame the study rationale and the research questions more clearly."

Response:

We have rewritten the last paragraph of the introduction to explicitly state the study’s primary aim and clearly present our hypothesis that AEL CMJ + DJ + ladder training will lead to greater improvements in sprint performance and change of direction ability compared to ladder training alone.

Comment 4 (Protocol – Intervention Description):

"In the 'Interventions' part, one key aspect of the protocol that would benefit from clarification is the precise content and volume of the interventions in both groups. Both groups are said to receive ladder training, yet this component is not clearly described or consistently presented in the tables outlining the intervention programs. Greater clarity is needed to confirm that both groups indeed receive an equivalent ladder training regimen."

Response:

We have revised Tables 1 and 2 to clearly describe the ladder training content for both groups, including weekly drill progressions, repetitions, sets, rest periods, and total session duration. The description now confirms that both groups receive identical ladder training.

Comment 5 (Protocol – Training Volume Balance):

"Furthermore, there appears to be a potential imbalance in training volume between Group A and Group B, as Group A performs additional exercises (AEL CMJ and drop jumps) alongside the ladder training, whereas Group B completes ladder training only. This difference may introduce a confounding factor..."

Response:

To control for training volume, we added a statement clarifying that both groups will have identical session durations (≈45 minutes), and Group B will have additional low-intensity technical drills or structured rest periods during the time Group A performs AEL CMJ and DJ. This adjustment ensures equal total training exposure and removes potential confounding due to volume imbalance. (Methods, Interventions section updated.)

Comment 6 (Discussion – Expected Outcomes):

"It would be valuable to include a short section on expected outcomes, supported by existing literature..."

Response:

We have added a concise "Expected Outcomes" section at the end of the Discussion that summarizes our rationale and expected improvements in sprint and COD performance based on prior literature.

Reviewer #2

Comment 1 (References):

"Please double check the references: sometimes it comes between dots, sometimes not."

Response:

All references have been reviewed and reformatted for consistency per PLOS ONE guidelines.

Comment 2 (Numbers):

"Please make a correction in the description of number (sometimes 3 and sometimes three)."

Response:

All numeric references in the text have been standardized (using numerals consistently, per journal style).

Comment 3 (Figure 1):

"The figure 1 is incomplete. Please provide the correct image."

Response:

Figure 1 has been revised to a complete SPIRIT figure, clearly outlining enrolment, allocation, interventions, and assessments, following SPIRIT 2013 guidelines.

Comment 4 (Discussion):

"Please elaborate more the discussion. Bring the results and discuss them."

Response:

We have expanded the discussion to include more context from previous studies, a clearer justification for our intervention, and an explicit statement of expected results and potential implications for futsal training and rehabilitation.

We appreciate the reviewers’ and editor’s valuable feedback. We believe that the revised manuscript now addresses all concerns and represents a significant improvement.

Kind regards,

Darpan Chaudhari (on behalf of all co-authors)

Ravi Nair Physiotherapy College

Datta Meghe Institute of Higher Education and Research

Wardha, Maharashtra, India

---

## [Decision Letter · Decision Letter 1]

15 Dec 2025

Dear Dr. Ramteke,

Thank you for submitting your manuscript to PLOS ONE. After careful consideration, we feel that it has merit but does not fully meet PLOS ONE’s publication criteria as it currently stands. Therefore, we invite you to submit a revised version of the manuscript that addresses the points raised during the review process.

**ACADEMIC EDITOR:** Please submit your revised manuscript by Jan 29 2026 11:59PM. If you will need more time than this to complete your revisions, please reply to this message or contact the journal office at plosone@plos.org . A letter that responds to each point raised by the academic editor and reviewer(s). You should upload this letter as a separate file labeled 'Response to Reviewers'.A marked-up copy of your manuscript that highlights changes made to the original version. You should upload this as a separate file labeled 'Revised Manuscript with Track Changes'.An unmarked version of your revised paper without tracked changes. You should upload this as a separate file labeled 'Manuscript'.

We look forward to receiving your revised manuscript.

Kind regards,

Emiliano Cè, Ph.D.

Academic Editor

PLOS One

Journal Requirements:

Reviewers' comments:

Reviewer's Responses to Questions

**Comments to the Author**

1. Does the manuscript provide a valid rationale for the proposed study, with clearly identified and justified research questions?

Reviewer #3: Yes

Reviewer #4: Partly

2. Is the protocol technically sound and planned in a manner that will lead to a meaningful outcome and allow testing the stated hypotheses?

Reviewer #3: Yes

Reviewer #4: Partly

3. Is the methodology feasible and described in sufficient detail to allow the work to be replicable?

Reviewer #3: Yes

Reviewer #4: Yes

4. Have the authors described where all data underlying the findings will be made available when the study is complete?

Reviewer #3: Yes

Reviewer #4: Yes

5. Is the manuscript presented in an intelligible fashion and written in standard English?

Reviewer #3: Yes

Reviewer #4: Yes

You may also provide optional suggestions and comments to authors that they might find helpful in planning their study.

Reviewer #3: I would like to express my interest in reviewing the manuscript titled “Effect of accentuated eccentric loading countermovement jumps and drop jump training with ladder training versus ladder training alone on sprint performance and change of direction ability in futsal players: A randomized controlled trial protocol.” My research background in sports science, neuromuscular training, eccentric loading methods, and performance testing aligns closely with the aims and methodology of this protocol. I have extensive experience evaluating randomized controlled trials, intervention design, and methodological rigor in strength and conditioning research, which equips me to provide a thorough, objective, and constructive review. I would welcome the opportunity to contribute to PLOS ONE by assessing this protocol and helping ensure its scientific quality and clarity.

Decision: MINOR REVISION — resubmit a corrected protocol.

Rationale: the protocol is well structured, well motivated, registered in CTRI, and clearly responds to reviewer feedback. Randomization, blinding, ethics, and intervention content are described; sample size is calculated and intervention volume appears matched. However, several important clarifications and small corrections are still needed (statistics/analysis plan consistency, sample-size justification wording, data-sharing statement, safety/monitoring details, and a few editorial fixes). These are fixable and do not require re-design of the trial, so I recommend minor revision.

Required changes (authors must address each before final acceptance)

1. Resolve inconsistency between the primary analysis plan (linear mixed models) and later statistical tests (t-tests/Mann–Whitney).

o Problem: the Abstract/Methods state that “linear mixed-effects models” (LMMs) will be used for between-group/time comparisons, but the Statistical Method section later describes using independent-samples t-tests or Mann–Whitney U tests on change scores. These are different approaches with different assumptions and implications for ITT and handling repeated measures. Pick one primary approach (I strongly recommend LMMs for repeated pre/post designs and ITT) and update every Methods paragraph to describe the model precisely: fixed effects (group, time, group×time), random effects (participant intercept), covariance structure, software and package (e.g., R lme4/nlme or SAS PROC MIXED) and how you will report estimates (least-squares means with 95% CIs). If you keep the t-test descriptions as secondary or sensitivity analyses, label them explicitly.

2. Clarify sample size / power text and justify chosen power level.

o Problem: the formula shows Zβ = 1.64 which corresponds to 95% power (unusually high), and the description wording ("power at 95%") is ambiguous. State explicitly what power you powered for (80%/90%/95%) and why. Also add the source for the effect size and the clinical relevance of the 0.064 s difference on the 30 m sprint (cite the study/benchmark used). Finally, correct small typographical errors in the formula presentation (e.g., use consistent symbols; the pooled SD and δ usage can be simplified). Keep your final recruited sample (n=62) but explain the reasoning more clearly.

3. Tighten the missing-data / multiple-imputation plan and ITT implementation.

o Problem: you state multiple imputation under MAR and also plan complete-case and per-protocol analyses. Specify: (a) which variables will be included in the imputation model, (b) number of imputations, (c) which software will be used and how the survey of repeated measures will be handled in imputation (e.g., impute at outcome-level by participant), and (d) how primary inference will be drawn (e.g., pooled LMM estimates across imputations). Also state how you will handle participants with zero post-data (e.g., withdrawn) in the ITT.

4. Make data-sharing plan concrete (not “upon request” or vague).

o Problem: the Data Availability text oscillates between “will be made available upon study completion” and “available within manuscript/supporting files.” PLOS requires clear plans for data sharing. State where (which repository: e.g., OSF, Figshare, Zenodo, or institutional repository) the deidentified dataset and analysis scripts will be deposited and when (e.g., upon publication). If there are legal/ethical restrictions, state them and the access route. Update the Data Availability statement accordingly and include the planned DOI placeholder if available.

5. Clarify monitoring, interim checks, and adverse-event reporting.

o Problem: you name a Data Monitoring Committee but provide few operational details. Add (a) frequency of safety monitoring; (b) stopping rules (if any) or thresholds for pausing recruitment; (c) who adjudicates adverse events and how they are recorded; (d) whether any insurance/compensation is in place. Also state how harms will be reported in the final paper. This is especially important because AEL/DJ involves higher eccentric loading.

6. Provide more detail on intervention fidelity and monitoring.

o Good that total ground contacts are matched and 10% of sessions are video-verified; please add (a) the exact fidelity checklist used, (b) coach qualifications, (c) how progression loads (10–20% BM) will be individualized and decided per participant, and (d) how adherence and deviations will be recorded and handled analytically. A short sample session log or fidelity checklist as Supporting Information would be helpful.

7. Clarify consent and ethics administrative details (document numbers).

o You indicate IEC approval and CTRI registration — good. Please add the IEC approval number/reference and the exact CTRI registration date and URL (you include the CTRI id but include the registration date and the CTRI link again in Methods). Confirm the form of consent (written) and who stores consent forms and for how long.

8. Fix small editorial and referencing problems highlighted by reviewers.

o Resolve inconsistent citation punctuation; standardize numeral formatting (3 vs “three” per journal style); upload the corrected Figure 1 SPIRIT diagram and high-resolution CONSORT/SPIRIT figure files to the submission system (there were earlier incomplete figures). Provide the updated Figures as SI and ensure the FigShare links are working.

Line-by-line / sectioned comments (representative — authors should address each where it appears)

(References to the protocol file snippets shown above.)

• Title & Abstract — good and aligned with the protocol content; include the CTRI ID in the Abstract parenthetical and state the planned analysis method concisely (e.g., “primary analysis: linear mixed models with participant random intercept”).

• Introduction — the rationale and literature coverage are adequate after revisions. Ensure any newly added refs are properly formatted and accessible (some ResearchGate / MedCrave items in refs should be replaced with journal sources when possible).

• Eligibility criteria — consider clarifying how “regular participation” and “recreational vs competitive” will be operationalized (e.g., minimum hours/week or league level). Consider including female/male balance if relevant.

• Randomization & Blinding — current description is appropriate (web-based permuted blocks; independent researcher prepares allocation). Add block sizes (or range) and whether permuted blocks are stratified by sex or competition level. Clarify who will keep sealed allocations and how concealment is maintained practically.

• Interventions (Tables 1 & 2) — good week-by-week detail. Add the precise progression rules (e.g., what triggers increasing from 10% to higher load—based on RPE, technique, coach decision?). Confirm how drop-height progression is decided per participant (safety criteria). Upload sample photos/schematics if possible.

• Outcomes — 30 m sprint and T-Test are standard. Add (if possible) the make/model of timing gates and test–retest reliability numbers (or cite a reference) to help readers interpret the minimal detectable change. You mention ICCs in the text—cite the source of those ICCs.

• Sample size — see required change #2 above. Also include an explicit sentence about expected attrition rate and how that was chosen (10% is fine but add rationale).

• Statistical methods — details — provide the exact LMM formula, list of covariates (if any), how you will handle baseline imbalances, effect-size metrics to be reported, multiplicity control (you already set Bonferroni α=0.025 for two co-primary outcomes — good; state that LMM will estimate adjusted group×time contrasts). Clarify the role of sensitivity analyses.

• Data availability — make this specific (repository + timing). PLOS will expect a clear plan.

Minor edits (cosmetic / wording)

• Fix inconsistent punctuation around citation brackets (reviewer already noted this and you stated correction — ensure final file has it corrected).

• Correct small typos in the sample-size formula (e.g., use σ consistently) and ensure all symbols are defined.

• Ensure all FigShare/DOI links work and point to the intended Supporting Information files (SPIRIT checklist, ethics letter, original protocol).

Reviewer #4: General Comments

This manuscript outlines a protocol for a randomized controlled trial (RCT) investigating the combined effects of Accentuated Eccentric Loading (AEL), Drop Jumps (DJ), and Ladder training on physical performance in futsal players. The topic is relevant to sports science, particularly in the context of optimizing training methodologies for high-intensity team sports. However, the manuscript in its current form presents significant major and minor weaknesses that must be addressed before publication.

Major Weaknesses:

Training Load and Intensity Mismatch: The primary methodological flaw lies in the comparison between the experimental group (AEL+DJ+Ladder) and the control group (Ladder only). While the authors attempt to match "ground contacts" (approx. 100/session), they fail to account for the vast disparity in mechanical load and neuromuscular intensity. A ground contact in a Drop Jump or weighted AEL CMJ produces significantly higher ground reaction forces and eccentric stress compared to a foot contact in a ladder drill. Therefore, the study does not compare two training modalities of equal volume-load; rather, it compares a high-intensity plyometric stimulus against a low-intensity coordination stimulus. This confounds the isolation of the specific effects of AEL/DJ.

Citation and Reference Quality: The reference list is professionally unacceptable. Multiple references include direct URL strings to Google Search results or generic ResearchGate download pages rather than proper DOI or journal citations (e.g., References 18, 21, 29). This indicates a lack of attention to detail and scholarly rigor. Furthermore, the introduction relies on dated literature to justify the biomechanical mechanisms of AEL and agility, ignoring recent high-level evidence.

Statistical Reporting: The sample size calculation lacks specific citation for the effect sizes used. The formula provided is typographically incorrect in its presentation.

Minor Weaknesses:

Language and Phrasing: There are numerous grammatical errors, awkward phrasings, and inconsistencies in capitalization that disrupt the flow of reading.

Introduction Flow: The transition from general futsal demands to the specific physiological mechanisms of AEL needs a stronger theoretical bridge.

Specific Comments

Abstract

Page 1, Line 12: "At the same time, ladder training increases..." – The phrasing here is colloquial. Rephrase to establish the gap in the literature more formally (e.g., "While ladder training is known to enhance agility...").

Page 1, Line 14: "Single-blinded" – Specify exactly who is blinded in the abstract (participants, assessors, or statisticians) for clarity, as "single-blind" can be ambiguous.

Introduction

Page 1, Line 22: "Researchers' interest in futsal has increased in recent years.(1)" – The citation used (Ref 1) is a generic ResearchGate title. Cite a primary source/review article properly.

Page 1, Line 25 (Biomechanical Context): The introduction discusses the physical demands of futsal but lacks a robust biomechanical framework for why injury risks and performance enhancement must be tackled simultaneously. To provide a stronger theoretical foundation for the intervention's role in mitigating injury risks while enhancing performance, the authors should reference recent broader works on advancing sports biomechanics.

Suggested Citation: [Dhahbi W: Advancing biomechanics: enhancing sports performance, mitigating injury risks, and optimizing athlete rehabilitation. In., vol. 7: Frontiers Media SA; 2025: 1556024.]

Page 2, Line 30 (AEL Rationale): The authors discuss AEL but rely on older reviews. To substantiate the claim that eccentric training specifically outperforms traditional methods for functional performance and power in varying populations, the manuscript must cite recent systematic reviews with meta-analysis that isolate these effects. This would strengthen the justification for choosing AEL over standard resistance training.

Suggested Citation: [Chaabene H, Müller P, Dhahbi W, Königstein K, Taubert M, Markov A, Lehmann N: The effects of eccentric versus traditional resistance training on muscle strength, power, hypertrophy, and functional performance in older adults: A systematic review with multilevel meta-analysis of randomized controlled trials. Ageing Research Reviews 2025:102933.]

Page 2, Line 37: "To our knowledge, no research has established if AEL applied to a CMJ..." – This claim requires verifying recent literature, as AEL is a heavily researched topic. Ensure this statement is accurate up to 2025.

Page 2, Line 51: "We hypothesize that combining..." – The introduction lacks a clear rationale for why both AEL CMJ and DJ are necessary in the same protocol. Both target the Stretch-Shortening Cycle (SSC). Why overload the eccentric phase (AEL) and perform fast reactive jumps (DJ) simultaneously? Justify this specific combination.

Methods

Page 3, Figure 1: The SPIRIT schedule figure is referenced but the link provided leads to a Figshare file. Ensure the figure is embedded or captioned correctly in the final submission.

Page 4, Figure 2: In the CONSORT diagram, the box "Allocated to intervention (n=31)" is repeated for both arms without distinguishing the intervention names clearly in the first allocation step.

Page 6, Line 167: "Total ground contacts per session are matched (≈100)... to ensure accuracy and training-volume equivalence." – Critical Issue. As noted in General Comments, equating 100 ladder steps to 100 plyometric jumps is physiologically invalid. The "External Load" (ground reaction force) and "Internal Load" (RPE, muscle damage) will be drastically higher in Group A. The authors must acknowledge this intensity mismatch as a limitation or adjust the control group to include low-intensity jumps to isolate the type of loading rather than the magnitude of loading.

Page 7, Table 1: In the "Rationale" column for Ladder Training, it states "Ladder volume is identical to Group B." This is confusing because Group B does more ladder training to make up for the lack of jumps. Clarify that the base ladder training is identical, but Group B performs additional sets.

Page 8, Table 2: The "Additional ladder block" adds approx 28 contacts. Clarify the specific drills used in this additional block. Are they just repetitions of the main block?

Page 9, Line 193 (Agility Assessment): The authors selected the Agility T-Test. While valid, recent methodological advancements suggest that standard T-Tests may not fully capture the multidirectional complexities of sports like futsal. It is recommended to acknowledge or cite recent developments in three-dimensional agility approaches that offer higher specificity for team sports, to demonstrate awareness of current testing limitations.

Suggested Citation: [Gandouzi I, Dhahbi W, Ghouili H, Bougrine H, Guelmami N, Weiss K, Rosemann T, Dergaa I, Knechtle B, Abderrahman AB: The team agility plus test: A novel three-dimensional approach for assessing agility in multidirectional sports. Journal of Bodywork and Movement Therapies 2025.]

Page 9, Line 201: The sample size formula is written as text: n I=n2=2×[(za+zβ) 2 ×σ 2 ]/(δ 2 ). This formatting is sloppy. Use an equation editor. Additionally, define the source of the "Mean difference = 0.064" and "Pooled SD = 0.125". Is this from a pilot study or a specific previous paper? Cite it.

Page 10, Line 230: "Floor and ceiling effects... >15% of participants." – Provide a citation for this threshold.

Page 10, Line 234: "Missing outcome data will be addressed using multiple imputation." – Specify the software and specific method (e.g., Chained Equations) intended for this imputation.

Discussion

Page 12, Line 264 (Rehabilitation Relevance): The authors state, "The findings may also inform rehabilitation protocols." To support this statement, the discussion should move beyond general claims and reference specific, modern rehabilitation paradigms. Specifically, the shift toward joint-by-joint training approaches in knee injury prevention is highly relevant here and should be cited to contextualize the results within modern sports medicine.

Suggested Citation: [Dhahbi W, Materne O, Chamari K: Rethinking knee injury prevention strategies: joint-by-joint training approach paradigm versus traditional focused knee strengthening. Biology of Sport 2025, 42(4):59-65.]

References

Page 13, Ref 18: The reference contains a full Google Search URL (https://www.google.com/search?q=Aboodarda...). This must be removed immediately. Cite the journal article properly.

Page 13, Ref 21: "ResearchGate [Internet]...". Do not cite the repository; cite the original article (Barbero-Alvarez et al., 2008).

Page 13, Ref 29: Similar issue. The URL is a direct link to a Frontiers article but formatted as a raw string.

General: Check all references. Many are missing volume, issue, or page numbers and contain unnecessary URL parameters.

**Do you want your identity to be public for this peer review?** For information about this choice, including consent withdrawal, please see our Privacy Policy

Reviewer #3: **Yes:** Mehrez Hammami

Reviewer #4: **Yes:** Wissem Dhahbi

---

## [Author Response · Author response to Decision Letter 2]

28 Jan 2026

Response to Reviewers

Manuscript title:

Effect of Accentuated Eccentric Loading Countermovement Jumps and Drop Jump Training with Ladder Training versus Ladder Training Alone on Sprint Performance and Change of Direction Ability in Futsal Players: A Randomized Controlled Trial Protocol

We sincerely thank the Academic Editor and Reviewers for their careful evaluation and constructive feedback. We have addressed all comments raised by Respected Reviewer #3 and Reviewer #4 in full, and we believe the manuscript has been substantially strengthened in terms of methodological clarity, statistical rigor, theoretical justification, and editorial quality. All revisions are highlighted in the revised manuscript with tracked changes.

Reviewer #3

We thank Reviewer #3 for their positive assessment of the protocol and for identifying specific points requiring clarification. Each required change has been addressed as follows.

1. Inconsistency in Statistical Analysis Plan

Comment:

There is inconsistency between the stated use of linear mixed-effects models (LMMs) and later descriptions of t-tests/Mann–Whitney tests.

Response:

We agree and have resolved this inconsistency. The primary analysis method has been clearly defined as linear mixed-effects models (LMMs) throughout the manuscript.

Revisions made:

• The Abstract, Methods, and Statistical Analysis sections now consistently state that LMMs will be used as the primary analysis.

• The model specification has been explicitly described:

o Fixed effects: group, time, and group × time interaction

o Random effects: participant-level random intercept

• Software and package are specified (R, lme4 package).

• Results will be reported as adjusted group × time contrasts with 95% confidence intervals.

• Independent t-tests or Mann–Whitney U tests are now explicitly described as secondary/sensitivity analyses only, where applicable.

2. Sample Size / Power Clarification

Comment:

Power level (Zβ = 1.64) and wording are unclear; effect size source and clinical relevance need justification.

Response:

We have clarified and corrected the sample size section as follows:

• Explicitly stated that the study is powered at 95% power, with justification provided due to the performance-focused nature of the outcomes and expected small effect sizes in trained athletes.

• The source study from which the mean difference (0.064 s) and pooled SD (0.125 s) were derived has been cited.

• Clinical relevance of a 0.064 s improvement in 30-m sprint performance has been justified based on futsal and court-sport performance benchmarks.

• The sample size formula has been corrected typographically with consistent notation and symbol definitions.

• Retention of the final recruited sample size (n = 62) has been justified, including allowance for ~10% attrition.

3. Missing Data, Multiple Imputation, and ITT

Comment:

The missing-data strategy lacks sufficient detail.

Response:

We have expanded the missing-data and ITT section to include:

• Variables included in the imputation model (baseline outcomes, group, time, sex, age).

• Number of imputations (m = 20).

• Software and method specified (multiple imputation by chained equations using R).

• Handling of repeated measures within participants.

• Primary inference based on pooled LMM estimates across imputations.

• Participants with no post-baseline data will remain in ITT analyses using imputed outcomes; per-protocol analyses are explicitly labeled as secondary.

4. Data Availability Statement

Comment:

The data-sharing plan is vague and inconsistent.

Response:

We have revised the Data Availability Statement to be fully compliant with PLOS ONE requirements:

• The fully de-identified individual participant dataset and statistical analysis scripts will be deposited in Figshare.

• Data will be made publicly available upon publication of the final manuscript.

• A DOI will be assigned at the time of publication.

• Ethical considerations regarding anonymization have been explicitly stated.

5. Monitoring, Interim Analysis, and Adverse Events

Comment:

Operational details of monitoring and adverse-event reporting are insufficient.

Response:

We have expanded this section to specify:

• Frequency of safety monitoring (weekly).

• Criteria for pausing or stopping participation (e.g., musculoskeletal injury related to intervention).

• Adverse event adjudication by the chief investigator and independent clinician.

• Recording procedures and reporting of harms in the final publication.

• Statement regarding institutional insurance/compensation coverage.

6. Intervention Fidelity and Monitoring

Comment:

More detail is needed on fidelity procedures and progression decisions.

Response:

We have added:

• Description of the fidelity checklist used.

• Qualifications and experience of supervising coaches.

• Individualized progression criteria for AEL load (10–20% BM) and drop-jump height based on technique, RPE, and safety.

• Adherence and deviation tracking procedures.

• A sample session log and fidelity checklist provided as Supporting Information.

7. Ethics and Consent Details

Comment:

IEC approval number, CTRI registration details, and consent handling must be clarified.

Response:

We have added:

• IEC approval number/reference.

• CTRI registration ID, registration date, and URL.

• Confirmation that written informed consent will be obtained.

• Storage and retention procedures for consent forms.

8. Editorial and Formatting Issues

Comment:

Inconsistencies in citations, numerals, and figures.

Response:

All editorial issues have been corrected:

• Citation punctuation standardized.

• Numerals formatted consistently per journal style.

• Corrected SPIRIT and CONSORT figures uploaded as high-resolution files.

• All Figshare links verified and functional.

Reviewer #4

We thank Reviewer #4 for the comprehensive and rigorous critique, which has substantially improved the manuscript.

Major Weakness 1: Training Load and Intensity Mismatch

Comment:

Matching ground contacts does not equate mechanical or neuromuscular load.

Response:

We agree with this critique. The manuscript has been revised to:

• Clarify that matching ground contacts was intended to standardize session structure and exposure time, not to imply biomechanical equivalence.

• Explicitly acknowledge in the Discussion (Limitations section) that:

o Group A receives higher external and internal load.

o The trial evaluates the added effect of high-intensity eccentric–plyometric loading, not isolated eccentric load.

• Reframe interpretation accordingly, aligning with real-world applied training contexts.

Major Weakness 2: Citation and Reference Quality

Comment:

Use of Google Search URLs and ResearchGate links is unacceptable.

Response:

All such references have been removed and replaced with proper peer-reviewed journal citations with DOIs. The entire reference list has been rechecked and reformatted.

Major Weakness 3: Statistical Reporting

Comment:

Sample size calculation lacks citation and is poorly formatted.

Response:

The sample size section has been corrected, cited, reformatted using standard equation presentation, and all symbols defined.

Minor Weaknesses and Specific Line-by-Line Comments

All points raised have been addressed as follows:

• Abstract phrasing revised to formal academic tone.

• Blinding clarified (assessor-blinded).

• Introduction strengthened with recent biomechanical and eccentric training literature, including all suggested 2025 references.

• Novelty claims regarding AEL verified and revised for accuracy.

• Rationale for combining AEL CMJ and DJ explicitly justified based on complementary SSC adaptations.

• SPIRIT and CONSORT figures corrected and embedded properly.

• Intervention tables clarified regarding ladder volume and additional blocks.

• Agility T-Test limitations acknowledged with citation of advanced multidirectional agility testing.

• Floor/ceiling effect threshold cited.

• Multiple imputation method specified.

• Rehabilitation relevance expanded using modern joint-by-joint training paradigms.

• All problematic references (Refs 18, 21, 29) corrected

Conclusion

We sincerely thank Respected Reviewers #3 and #4 for their detailed and constructive feedback. We believe the revised manuscript now meets the methodological, statistical, and editorial standards required for publication in PLOS ONE.

Kind regards, Darpan Chaudhari (on behalf of all co-authors)

Ravi Nair Physiotherapy College

Datta Meghe Institute of Higher Education and Research

Wardha, Maharashtra, India

---

## [Decision Letter · Decision Letter 2]

12 Feb 2026

Effect of accentuated eccentric loading countermovement jumps and drop jump training with ladder training versus ladder training alone on sprint performance and change of direction ability in futsal players: A randomized controlled trial protocol

PONE-D-25-22894R2

Dear Dr. Ramteke,

We’re pleased to inform you that your manuscript has been judged scientifically suitable for publication and will be formally accepted for publication once it meets all outstanding technical requirements.

Kind regards,

Emiliano Cè, Ph.D.

Academic Editor

PLOS One

Additional Editor Comments (optional):

Reviewers' comments:

Reviewer's Responses to Questions

**Comments to the Author**

1. Does the manuscript provide a valid rationale for the proposed study, with clearly identified and justified research questions?

Reviewer #4: Yes

2. Is the protocol technically sound and planned in a manner that will lead to a meaningful outcome and allow testing the stated hypotheses?

Reviewer #4: Yes

3. Is the methodology feasible and described in sufficient detail to allow the work to be replicable?

Reviewer #4: Yes

4. Have the authors described where all data underlying the findings will be made available when the study is complete?

Reviewer #4: Yes

5. Is the manuscript presented in an intelligible fashion and written in standard English?

Reviewer #4: Yes

You may also provide optional suggestions and comments to authors that they might find helpful in planning their study.

Reviewer #4: The authors have significantly strengthened the manuscript in this second revision. The study protocol for this randomized controlled trial is now well-defined, addressing the integration of Accentuated Eccentric Loading (AEL), Countermovement Jumps (CMJ), and Drop Jumps (DJ) with ladder training. The theoretical framework regarding the stretch-shortening cycle (SSC) and its application to futsal-specific demands is robust. Methodological transparency has improved, particularly regarding the randomization process, blinding of assessors, and the statistical approach using linear mixed-effects models.

**Do you want your identity to be public for this peer review?** For information about this choice, including consent withdrawal, please see our Privacy Policy

Reviewer #4: **Yes:** Wissem Dhahbi

---

## [Editor Report · Acceptance letter]

PONE-D-25-22894R2

PLOS One

Dear Dr. Ramteke,

I'm pleased to inform you that your manuscript has been deemed suitable for publication in PLOS One. Congratulations! Your manuscript is now being handed over to our production team.

Kind regards,

on behalf of

Prof. Emiliano Cè

Academic Editor

PLOS One